# Psychometric Performance of a Condition-Specific Quality-of-Life Instrument for Dutch Children Born with Esophageal Atresia

**DOI:** 10.3390/children9101508

**Published:** 2022-10-01

**Authors:** Chantal A. ten Kate, Hanneke IJsselstijn, Michaela Dellenmark-Blom, E. Sofie van Tuyll van Serooskerken, Maja Joosten, René M. H. Wijnen, Michiel P. van Wijk

**Affiliations:** 1Department of Pediatric Surgery and Intensive Care Children, Erasmus University Medical Centre-Sophia Children’s Hospital, 3015 CN Rotterdam, The Netherlands; 2Department of Pediatric Surgery, Queen Silvia Children’s Hospital, Sahlgrenska University Hospital, 41650 Gothenburg, Sweden; 3Department of Pediatric Surgery, Wilhelmina Children’s Hospital, University Medical Center Utrecht, 3584 EA Utrecht, The Netherlands; 4Department of Pediatric Surgery, Radboud University Medical Center, Amalia Children’s Hospital, 6525 GA Nijmegen, The Netherlands; 5Department of Pediatric Gastroenterology and Nutrition, Amsterdam UMC–Emma Children’s Hospital, University of Amsterdam, 1105 AZ Amsterdam, The Netherlands

**Keywords:** feasibility, validity, reliability, questionnaire, disease-specific

## Abstract

A condition-specific instrument (EA-QOL©) to assess quality of life of children born with esophageal atresia (EA) was developed in Sweden and Germany. Before implementing this in the Netherlands, we evaluated its psychometric performance in Dutch children. After Swedish–Dutch translation, cognitive debriefing was conducted with a subset of EA patients and their parents. Next, feasibility, reliability, and validity were evaluated in a nationwide field test. Cognitive debriefing confirmed the predefined concepts, although some questions were not generally applicable. Feasibility was poor to moderate. In 2-to-7-year-old children, 8/17 items had >5% missing values. In 8-to-17-year-old children, this concerned 3/24 items of the proxy-report and 5/14 items of the self-report. The internal reliability was good. The retest reliability showed good correlation. The comparison reliability between self-reports and proxy-reports was strong. The construct validity was discriminative. The convergent validity was strong for the 2-to-7-year-old proxy-report, and weak to moderate for the 8-to-17-year-old proxy-report and self-report. In conclusion, the Dutch-translated EA-QOL questionnaires showed good reliability and validity. Feasibility was likely affected by items not deemed applicable to an individual child’s situation. Computer adaptive testing could be a potential solution to customizing the questionnaire to the individual patient. Furthermore, cross-cultural validation studies and implementation-evaluation studies in different countries are needed.

## 1. Introduction

As more newborns with esophageal atresia (EA) survive, long-term morbidities and quality of life become more relevant. Reported morbidities include gastrointestinal and pulmonary problems (e.g., dysphagia, gastroesophageal reflux, and/or recurrent airway infections) [1,2], growth retardation [3], reduced exercise capacity [4], and impaired motor function [5]. A significant long-term aspect is the burden of disease, reflected in patient-reported outcome measures (PROMs) on health status (HS) and quality of life (QoL). HS describes a patient’s well-being in terms of functioning [6], while QoL focuses on perception about one’s functioning [7]. In the literature, these two concepts are often combined to health-related QoL (HRQoL) [8].

Despite inconsistent findings, studies indicated that children with EA report lower generic HRQoL than do healthy children [9,10,11,12]. Condition-specific instruments tend to be more sensitive to the detection and discrimination of clinical morbidities, and more suitable for assessing disease burden [13]. For that reason, Dellenmark-Blom and coworkers have developed a condition-specific instrument to measure HRQoL in children with EA: the EA-QOL© questionnaires [14,15,16]. This set of age-specific questionnaires covers multiple domains (see Appendix A) identified through focus group interviews with children and/or their parents [16]. The results of the protocolized validation process have been published for Swedish, German, Turkish, and Polish children [14,17,18].

Assessing HRQoL benefits health care for chronic conditions and enhances communication between children and professionals [19,20]. Therefore, generic questionnaires have been implemented in Dutch follow-up programs [21]. A condition-specific instrument could contribute to a better understanding of children’s perception of their disease impact, enabling tailor-made intervention strategies. However, given the heterogeneity of EA, its comorbidities, and follow-up strategies worldwide, the context in which an instrument would be implemented differs between countries. We assumed that the translated questionnaires would not necessarily frictionlessly fit our population. Before implementing the EA-QOL© questionnaire in clinical practice in the Netherlands, we evaluated its psychometric performance in Dutch children.

## 2. Methods

This is a cross-sectional study consisting of three phases (translation, cognitive debriefing, and field testing), similar to the other EA-QOL© validation studies and following international guidelines regarding the criteria for feasibility, reliability, and validity [14,17,18,22]. The study has been approved by the participating institutional review boards (IRB) (MEC-2019-0521, MEC-20-564/C, MEC-2020-6961 and MEC-2019-631). See Appendix A for a detailed description of methods and IRB-related data. 

### 2.1. Translation and Cognitive Debriefing

A Swedish–Dutch forward–backward translation was conducted [23] and reviewed by the Swedish developer to ensure conceptual equivalence. To ensure that all items were understood as intended by the Dutch target population [24], cognitive debriefing (Appendix A) was conducted in three groups, stratified by severity of complaints (Appendix A), as described previously [15]: (A) parents of 2-to-7-year-old children (proxy-report); (B1) parents of 8-to-17-year-old children (proxy-report); and (B2) 8-to-17-year-old children (self-report). Participants were interviewed face-to-face during an annual meeting of the Dutch patient support group in September 2019. Participants of groups B1 and B2 were related and interviewed separately and simultaneously. Participants filled out the questionnaire on paper while giving feedback on the clarity and adequacy of the items. The results were analyzed using content analysis. If necessary, we slightly rephrased instructions and items, after having consulted the Swedish developer. We obtained consensus on the final questionnaires for the field test.

### 2.2. Field Testing

A nationwide field test was conducted between August 2020 and April 2021 in the Netherlands. Participants without known intellectual disability who had sufficient command of the Dutch language were identified from the databases of four university hospitals that cover care for approximately 80% of the Dutch EA population. They were invited through a personal letter, containing a personal access code to fill out the questionnaires online (LimeSurvey GmbH version 2.06lts, Hamburg, Germany). Parents who were supposed to be the child’s primary care taker and/or children ≥8 years old filled out age-appropriate proxy-reports or self-reports of the EA-QOL© questionnaire [14] and the Pediatric Quality of Life Inventory™ 4.0 (PedsQL) questionnaire [25] (Appendix A). A general questionnaire on sociodemographic information and on digestive symptoms, feeding difficulties, and respiratory symptoms in the past four weeks was obtained as proxy-report in children <12 years old and as both self-report and proxy-report in children ≥12 years old. Parental educational level was classified according to the International Standard Classification of Education [26].

To examine the test–retest reliability, participants were invited to fill out the EA-QOL© questionnaire a second time three weeks after the initial response. If needed, reminders were sent twice maximally. The final reminder also included the questionnaire on paper with a pre-stamped envelope for reply.

The following data were retrieved from the patient records: sex, gestational age, birth weight, type of EA [27], presence of VACTERL (vertebral, anorectal, cardiac, tracheoesophageal, renal, and limb anomalies) association [28], type of primary surgery, postoperative complications (anastomotic leakage, pneumothorax, sepsis, wound infection, or recurrent fistula), history of gastrostomy, and history of esophageal dilatation. EA was considered long gap if staged repair had been performed. Small for gestational age was defined as birth weight <10th percentile [29]. Pneumothorax was defined as the need for a chest tube, sepsis as a positive blood culture, and wound infection according to the surgical site infection criteria of Centers for Disease Control and Prevention [30].

### 2.3. Statistical Analysis

Data are presented as number (%), median (interquartile range), or mean ± SD. Items were answered on a 5-point Likert scale and reversed linearly transferred to a 0–100 scale, with 100 as best possible score. Subscales and total scores were computed by the mean, with a maximum of 30% missing items per subscale. Items were described as mean ± SD (range). Feasibility (percentage of items with >5% missing values [15]) and psychometric criteria (skewness and kurtosis <2.0) were evaluated [31]. Feasibility was considered poor (>30%), moderate (10–30%), or good (<10%). Subscales and total scores were described as median (IQR) with floor and ceiling effects (percentage of respondents reporting, respectively, the minimum and maximum possible score <15%) [32].

Internal reliability was considered good if Cronbach’s alpha ≥0.7 for the scales [32]. External reliability—both proxy-self and test–retest comparison—was evaluated using intra-class coefficients (ICCs), using a two-way random model, single measures, and absolute agreement. It was considered poor (<0.50), moderate (0.50–0.74), good (0.75–0.90), or excellent (>0.90) [33].

Construct validity was determined through known-groups validity; Mann–Whitney U tests served to assess differences in total scores between clinical subgroups: patients with and without primary repair, with gastrostomy or ≥1 esophageal dilatation in history, with and without digestive symptoms, with feeding difficulties, and with respiratory symptoms in the past four weeks. We applied a Bonferroni correction to account for multiple comparison. As we assessed differences for 20 different variables, alpha was set at 0.05/20 = 0.0025. Effect sizes (ESs) were calculated by converting z-scores of the Mann–Whitney U tests (*r* = z/√n) [34] and were considered to strengthen the validity if moderate (>0.30) or large (>0.50). Children in clinical subgroups were hypothesized to have lower total scores.

Convergent validity was examined by correlating the proxy-reported and self-reported total scores with the concomitant PedsQL scores [25,35] using Spearman’s rho (*r*_s_), and concluded as poor (<0.40), moderate (0.40–0.59), good (0.60–0.79), or excellent (>0.80). Statistical analyses were performed using SPSS V.24.0 (IBM, Chicago, IL, USA), with a significance level of *p* < 0.05.

## 3. Results

### 3.1. Cognitive Debriefing

A review of the translations confirmed the intended conceptual content. Twenty-nine participants (19 parents and 10 children) were recruited for cognitive debriefing. Group A consisted of nine parents (11% male, age range 32–44 years) of children with mild (*n* = 2), moderate (*n* = 5), or severe (*n* = 2) complaints. Group B1 contained ten parents (30% male, age range 41–61 years), and group B2 contained ten children (40% male, age range 9–17 years) with mild (*n* = 4), moderate (*n* = 4) or severe (*n* = 2) complaints. 

Appendix A summarizes the cognitive debriefing results. Overall, participants understood the items correctly according to the predefined concepts. Parents considered two items (Can your child eat at the same pace as other children his/her age? & Does your child need to think of drinking a lot when he/she eats?) multi-interpretable. We rephrased those items as suggested. Although participants considered some items burdensome, none were rejected. Some items, e.g., questions on oral feeding in case of full dependency of (par)enteral feeding or questions on small stature while having physical height within normal ranges were repeatedly considered not applicable and unable to answer properly. To keep the translated version in line with the original, we did not adjust the response scale but modified the instructions. In the field test, participants were instructed to omit questions if not applicable. Some participants indicated that they had missed certain topics (Appendix A). To preserve the original structure of the questionnaire, we continued to the field test with the questionnaire in its current form.

### 3.2. Field Test

#### Study Population

In total, 101 parents of 2-to-7-year-old children, 136 parents of 8-to-17-year-old children, and 130 8-to-17-year-old children participated in the field test (response rate, respectively, 51%, 41%, and 39%, recruited nationwide [36]). Respectively, 26%, 38%, and 39% of them returned the questionnaire on paper Appendix A. The proportion of parents with high educational level was larger than that in the general population (58% vs. 36%) [37]. See Table 1 for demographic characteristics and Table 2 for clinical symptoms of the participants.

### 3.3. Item Evaluation

Feasibility was poor to moderate (Appendix A). Of the 2-to-7-year-old proxy-report, 8/17 items had >5% (6.9–32.7%) missing values, including all items of ‘Social isolation and stress’. Of the 8-to-17-year-old proxy-report, 5/24 items had >5% (5.8–28.7%) missing values. Of the 8-to-17-year-old self-report, 3/24 items had >5% (11.5–21.5%) missing values. Subscale and total scores are presented in Table 3. We did not observe any floor effects. Ceiling effects of >15% were found for ‘Social isolation and stress’ for the 2-to-7-year-old proxy-report, and for ‘Social relationships’, ‘Body perception’, and ‘Health and well-being’ of both the 8-to-17-year-old self-reports and proxy-reports.

### 3.4. Internal and External Reliability

Internal reliability was good for the total scores, but the Cronbach’s alpha for ‘Health and well-being’ was <0.7. For the proxy-self comparison, 128 child–parent couples were available, with good correlation for all subscales (Appendix A) and the total score (ICC 0.81). In the retest, 70 parents (69% of the original sample) of 2-to-7-year-old children, 82 parents (60%) of 8-to-17-year-old children, and 71 8-to-17-year-old children (55%) responded. Basic characteristics did not differ between respondents and non-respondents. Respectively, 6%, 17%, and 16% of the questionnaires were returned on paper. Clinical symptoms of none of the children differed from that in the initial test. Test–retest agreement was good for the total scores and most of the subscales (Table 3). Agreement was moderate for ‘Social isolation and stress’ in the 2-to-7-year-old proxy-report and ‘Social relationships’ and ‘Body perception’ in the 8-to-17-year-old self-report.

### 3.5. Construct Validity

Total scores of the 2-to-7-year-old proxy-report were lower for symptomatic children, with moderate to large ESs—except for children with heartburn, chest tightness, and airway infections. Total scores of the 8-to-17-year-old proxy-report were lower for children avoiding certain food, adjusting their portions, or increasing their fluid intake during meals, with moderate ESs. Total scores of the 8-to-17-year-old self-report were lower for children with dysphagia or dyspnea during physical activity and for children adjusting their portions or increasing their fluid intake during meals, with moderate ESs. See Table 4.

### 3.6. Convergent Validity

Total PedsQL scores showed a strong correlation with total EA-QOL© score of the 2-to-7-year-old proxy-report (*n* = 100, *r_s_* = 0.64, *p* < 0.001), a weak correlation with total score of the 8-to-17-year-old proxy-report (*n* = 135, *r_s_* = 0.39, *p* < 0.001), and a moderate correlation with the total score of the 8-to-17-year-old self-report (*n* = 130, *r_s_* = 0.54, *p* < 0.001). See Appendix A for complete subscale and total PedsQL scores (Appendix A).

## 4. Discussion

In this nationwide validation study of a condition-specific PROM for children with EA, we evaluated the psychometric performance of the Dutch-translated EA-QOL© questionnaires. Cognitive debriefing confirmed good understanding of the items according to the predefined concepts, but not all questions were deemed applicable for each individual child. Overall, the field test showed good internal and retest reliability for the total scores and most of the subscales. The construct validity was slightly discriminative. The convergent validity was variable, from weak to strong correlations.

In general, Dutch participants reported higher EA-QOL© scores than those in the Swedish–German validation study. From a clinical perspective, this could be explained by differing perceptions of symptoms—or perhaps fewer comorbidities. In our population, 2-to-7-year-olds tended to have fewer airway infections, and 8-to-17-year-olds had fewer complaints of heartburn and vomiting than in the Swedish–German study population. None of the Dutch children required parenteral nutrition in the field test, in contrast to 4 out of 124 Swedish–German children [14]. Given the response rates (39–51%), it may imply that patients with a larger symptom burden did not respond, which may influence the study findings. Considering the psychological distress of parenteral feeding [38], this difference could have contributed to the higher EA-QOL scores in the Dutch population.

Another clinical explanation is the potential influence of the COVID-19 pandemic. The second lockdown in the Netherlands overlapped with the field test period. Closure of primary and secondary schools for three and six months, respectively [39], significantly impacted children’s social life. Reduced social activities may have resulted in less negative confrontation with impairments of their chronic condition and leading to items being less applicable, while healthy children’s QoL was negatively affected by COVID-19 [40].

One could argue that the higher Dutch scores are related to test characteristics. Ceiling effects were present in both study populations, but floor effects (all <15%) were observed only in the Swedish–German population. However, validation of the well-established DISABKIDS, CHQ-CF87, and PedsQL instruments showed similar results, with rare floor effects but ceiling effects up to 86% [25,31,41]. Validation of the Dutch version of the CHQ-CF87, cross-culturally adapted from the United States, even showed no floor effects at all [41], like in our study. Moreover, the high-level child–parent agreements favor a clinical rather than a technical explanation for the differences between the Dutch and Swedish–German populations.

Next, the proportion of items with missing values in our study (up to 32.7%) was larger than that in previous studies [14,17], though missing values were not specifically reported for the Polish cohort [18]. Considering the cognitive debriefing results in our study (Appendix A), this was anticipated. We instructed participants in the field test to omit the questions they considered not applicable and noted that the omitted questions corresponded with those commented on during cognitive debriefing. Soyer and coworkers—who performed the Turkish field test of the EA-QOL© questionnaires in the outpatient clinic—did not share data on cognitive debriefing [17]. Rozensztrauch et al. reported a positive perception during in Polish cohort but did not specify who conducted the interviews [18]. Differences in study design could explain the abovementioned differences.

The wide—slightly skewed—age range within the groups could explain this poor feasibility. Toddlers’ perception of potential problems in daily functioning differs from that of school-aged children, and toddlers may be less capable of expressing their burden verbally. Moreover, not every toddler attends daycare, which may differ amongst countries. In the Netherlands, daycare attendance was even less during the COVID-19 pandemic. In a longitudinal cohort study, we showed that growth is slightly below the norm in younger children with EA but normalizes at 12 years [3]. This may explain the frequently omitted question on perception of having a short stature and emphasizes the need for cross-cultural adaptation of the questionnaire.

One could argue the rationale of examining the feasibility, reliability, and validity of an instrument for each country separately. However, our results support the current guidelines of PROM that this is essential before implementing a translated QoL questionnaire in research or clinical practice.

Differences in clinical presentation and follow-up care in different centers could impact the rating of a child’s QoL. Furthermore, one’s health perception might be subject to cultural differences between countries [42]. Culture is multi-aspect concept that requires further exploration in this context. For example, adequate coping skills may lead to positive illusionary bias [43] and, hence, to considering chronic healthcare problems and concomitant lifestyle factors such as dietary restrictions normal. This phenomenon as well as differential item functioning [44]—measuring different aspects in subgroups of participants due to perceptional differences—should be taken into account when implementing the EA-QOL© questionnaires in clinical practice and during cross-cultural evaluation.

Moreover, small sample sizes and heterogeneity (to which cross-cultural differences contribute) are known challenges for the soundness of PROMs in rare diseases. A possible solution might be computer adaptive testing (CAT), enabling customization of a questionnaire to an individual’s situation by using skip patterns that are based on the individual’s prior responses to administer items from an item bank [45]. For generic PROMs, the Patient-Reported Outcome Measure Information System contains item banks for physical, mental, and social health in adult and pediatric populations, selected from the literature and tested through various extensive item-response theory (IRT) models [46]. CATs have been developed to measure HRQoL in children with chronic conditions [47] but not in rare diseases such as EA. Generating an item bank for condition-specific items requires large sample sizes recruited from multiple countries [22]. Given the strong correlation of condition-specific scores with generic PedsQL scores, the added value of implementing the EA-QOL© questionnaires in clinical practice should first be established. A possible approach is to correlate scores to clinical outcomes, like has been carried out for the PedsQL and DUX-25 [48]. A next step could be to combine the internationally obtained validation results into an IRT model, using the original EA-QOL© items available before item reduction [16], with the addition of topics brought up during cognitive debriefing in multiple countries. However, further research in additional countries is needed to evaluate the potential of CAT for the EA-QOL© questionnaires in daily practice.

One of the strengths of this study is the relatively large sample size, considering that EA is a rare condition. Furthermore, participants were recruited nationwide. Some limitations should be addressed. We recruited participants from hospital databases and not only those who participated in follow-up programs. We did not collect data from non-participants; therefore, selection bias cannot be ruled out. Moreover, due to the relatively low response rates between 39 and 51% despite the two reminders we sent to non-responders, it remains possible that patients with the most severe symptoms and, therefore, worst QoL did not respond. Furthermore, the parental educational level was higher than in the general population. Although this is a common finding in the EA population [12,48] and in psychometric evaluation in general [49], it should be taken into account that this could lead to bias. Moreover, the online study set-up and some statistical assumptions differed from earlier EA-QOL© validation studies. Next, investigating sex-specific EA-QOL© scores was beyond the scope of this study. Still, it is recognized that females report lower QoL than do males [35,50]. In future cross-cultural evaluation, sex should therefore be considered as a potential confounder [51]. Lastly, the COVID-19 pandemic could have influenced the field test results.

## 5. Conclusions

The Dutch-translated EA-QOL© questionnaires showed good reliability and validity. Feasibility was most likely affected by items not deemed applicable to an individual child’s situation, as the cognitive debriefing made clear. Leading from this, CAT could be a potential solution to making the questionnaires more suitable for clinical practice in the Netherlands. Cross-cultural evaluation of the validation results obtained in multiple countries should further explore this.

## Figures and Tables

**Table 1 children-09-01508-t001:** Basic characteristics of the respondents, presented as *n* (%) or median (interquartile range). ISCED = International Classification of Education, EA = esophageal atresia, VACTERL = vertebral, anorectal, cardiac, tracheoesophageal, renal, or limb anomalies, ISCED = International Classification of Education [26]. ^A^ City with >100,000 citizens. ^B^ At time of filling out questionnaire. ^C^ Birth weight <10th percentile [29]. ^D^ According to Gross classification [27]. ^E^ According to Solomon criteria [28]. ^F^ gastric pull-up *n* = 1. ^F^ gastric pull-up *n* = 1, jejunal interposition *n* = 5.

	EAQOL 2–7 Years Proxy-Report (*n* = 101)	EAQOL 8–17 Years Proxy-Report (*n* = 136)	EAQOL 8–17 Years Self-Report (*n* = 130)
**Demographic characteristics**			
Region			
North	8 (7.9)	2 (1.5)	2 (1.5)
South	12 (11.9)	18 (13.2)	18 (13.8)
West	59 (58.4)	81 (59.6)	79 (60.8)
East	21 (20.8)	35 (25.7)	31 (23.8)
Foreign country	1 (1.0)	-	-
Urban area ^A^	25 (24.8)	35 (25.7)	33 (25.4)
**Parental characteristics**			
Age (years) ^B^	38.5 (35.4–41.7)	45.7 (41.9–49.4)	
Male	21 (20.8)	31 (22.8)	
Single caregiver	6 (5.9)	10 (7.4)	9 (6.9)
Born in the Netherlands	93 (92.1)	120 (88.2)	114 (87.7)
Parental educational level			
Low (ISCED 0–2)	7 (6.9)	18 (13.2)	18 (13.8)
Middle (ISCED 3–4)	28 (27.7)	46 (33.8)	43 (33.1)
High (ISCED 5–8)	66 (65.3)	72 (52.9)	68 (52.3)
Parent with chronic condition	8 (7.9)	13 (9.6)	13 (10.0)
**Child characteristics**			
Age (years) ^B^	5.0 (3.5–6.5)	13.6 (10.9–15.9)	13.8 (11.0–15.9)
Male	60 (59.4)	85 (62.5)	83 (63.8)
Gestational age (weeks)	37.7 (35.8–39.9)	38.0 (35.6–39.3)	38.0 (35.6–39.3)
Birth weight (grams)	2790 (1978–3300)	2740 (2200–3149)	2750 (2215–3200)
Preterm birth	36 (35.6)	47 (34.6)	46 (35.4)
Small for gestational age ^C^	38 (37.6)	49 (36.0)	46 (35.4)
Type of EA ^D^			
Type A	4 (4.0)	11 (8.1)	11 (8.5)
Type B	2 (2.0)	2 (1.5)	2 (1.5)
Type C	87 (86.1)	116 (85.3)	110 (84.6)
Type D	-	2 (1.5)	2 (1.5)
Type E	4 (4.0)	3 (2.2)	3 (2.3)
Unknown	4 (4.0)	2 (1.5)	2 (1.5)
Staged repair	8 (7.9)	14 (10.3)	14 (10.8)
VACTERL association ^E^	16 (15.8)	16 (11.8)	15 (11.5)
Type of repair			
Primary anastomosis	85 (84.2)	120 (88.2)	115 (88.5)
Delayed primary anastomosis	10 (9.9)	10 (7.4)	9 (6.9)
Esophageal replacement ^F^	1 (1.0) ^E^	6 (4.4) ^F^	6 (4.6) ^F^
Unknown	5 (5.0)	-	-
Postoperative complications			
Anastomotic leakage	15 (14.9)	16 (11.8)	16 (12.3)
Pneumothorax	32 (31.7)	46 (33.8)	43 (33.1)
Sepsis	13 (12.9)	11 (8.1)	11 (8.5)
Wound infection	7 (6.9)	5 (3.7)	5 (3.8)
Recurrent fistula	1 (1.0)	5 (3.7)	6 (4.6)
History of gastrostomy	15 (14.9)	18 (13.2)	17 (13.1)
History of ≥1 dilatation	53 (52.5)	68 (50.0)	66 (50.8)
Siblings	77 (76.2)	110 (80.9)	106 (81.5)

**Table 2 children-09-01508-t002:** Digestive symptoms, respiratory symptoms, and feeding difficulties in the four weeks prior to filling out the questionnaires, presented as *n* (%). ^A^ Only children ≥12 years old reported these items.

	EAQOL 2–7 Years Proxy-Report (*n* = 101)	EAQOL 8–17 Years Proxy-Report (*n* = 136)	EAQOL 8–17 Years Self-Report (*n* = 84) ^A^
**Symptoms**			
Heartburn	18 (17.8)	17 (12.5)	12 (14.3)
Vomiting during or after meals	21 (20.8)	6 (4.4)	2 (2.4)
Difficulty to swallow food	40 (39.6)	30 (22.1)	24 (28.6)
Food getting stuck	45 (44.6)	39 (28.7)	30 (35.7)
Complaints of pain while swallowing	16 (15.8)	9 (6.6)	8 (9.5)
Coughing	64 (63.4)	63 (46.3)	42 (50.0)
Wheezing	26 (25.7)	9 (6.6)	11 (13.1)
Dyspnea at rest	10 (9.9)	6 (4.4)	7 (8.3)
Dyspnea during physical activity	12 (11.9)	20 (14.7)	25 (29.8)
Chest tightness	3 (3.0)	9 (6.6)	14 (16.7)
Airway infections	14 (13.9)	12 (8.8)	3 (3.6)
Recurrent pulmonary problems	34 (33.7)	32 (23.5)	14 (16.7)
**Feeding difficulties**			
Avoiding food that is difficult to swallow	35 (34.7)	31 (22.8)	12 (14.3)
Eating small portions	43 (42.6)	26 (19.1)	13 (15.5)
Requiring energy-enriched food	21 (20.8)	8 (5.9)	5 (6.0)
Requiring adjusted food consistency	25 (24.8)	1 (0.7)	-
Needing >30 min to finish a meal	36 (35.6)	17 (12.5)	8 (9.5)
Requiring increased fluid intake	45 (44.6)	43 (31.6)	31 (36.9)
Nutrition through tube or gastrostomy	12 (11.9)	2 (1.5)	1 (1.2)
Receiving nutrition through infusion pump	-	-	-
Needing adult support while eating	28 (27.7)	4 (2.9)	14 (16.7)

**Table 3 children-09-01508-t003:** Descriptive values of the EA-QOL© questionnaires. SD = standard deviation, IQR = interquartile range, ICC = intra-class correlation coefficient, CI = confidence interval. ^A^ Digestive symptoms, feeding difficulties and respiratory symptoms in the 4 weeks prior to the retest did not differ from those in the 4 weeks prior to the initial test. ^B^ Median interval of 39 days (range 20–91). ^C^ Median interval of 50 days (range 20–138). ^D^ Median interval of 51 days (range 20–138).

	Items (*n*)	Respondents (*n*)	Median (IQR)	Floor, *n* (%)	Ceiling, *n* (%)	Cronbach’s Alpha	Respondents (*n*) ^A^	Level of Agreement, ICC (95% CI)
Children 2–7 years old (proxy-report)								
Eating	7	101	82.14 (65.48–92.86)	-	6 (5.9)	0.85	68	0.77 (0.65–0.85)
Physical health and treatment	6	92	75.00 (65.50–90.00)	-	9 (8.9)	0.79	62	0.84 (0.75–0.90)
Social isolation and stress	4	71	93.75 (68.75–100.00)	-	30 (29.7)	0.77	42	0.66 (0.44–0.0.80)
Total score	17	101	79.69 (66.99–91.07)	-	5 (5.0)	0.90	70 ^B^	0.86 (0.78–0.91)
Children 8–17 years old (proxy-report)								
Eating	8	128	81.70 (71.88–90.63)	-	18 (14.1)	0.74	76	0.74 (0.62–0.83)
Social relationships	7	134	89.29 (81.25–100.00)	-	44 (32.8)	0.75	80	0.81 (0.72–0.87)
Body perception	5	133	100.00 (85.00–100.00)	-	68 (51.1)	0.79	79	0.71 (0.56–0.81)
Health and well-being	4	134	91.67 (79.69–100.00)	-	37 (27.6)	0.59	80	0.80 (0.70–0.87)
Total score	24	136	87.50 (79.21–93.28)	-	6 (4.4)	0.86	82 ^C^	0.84 (0.77–0.90)
Children 8–17 years old (self-report)								
Eating	8	129	81.25 (68.75–92.19	-	17 (13.2)	0.69	69	0.78 (0.66–0.86)
Social relationships	7	129	89.29 (76.79–100.00)	-	37 (28.7)	0.73	70	0.68 (0.53–0.79)
Body perception	5	129	100.00 (90.00–100.00)	-	77 (59.7)	0.76	69	0.54 (0.35–0.69)
Health and well-being	4	128	89.58 (81.25–100.00)	-	38 (29.7)	0.39	68	0.70 (0.56–0.81)
Total score	24	130	86.71 (78.72–94.75)	-	8 (6.2)	0.85	71 ^D^	0.72 (0.85–0.81)

**Table 4 children-09-01508-t004:** (**a**) Comparison between clinical subgroups of total scores of the EA-QOL© questionnaire for children aged 2–7 years old (proxy-reports). Only patients for whom both clinical data and total scores EA-QOL© were available were included in the analyses. Asterisks indicate Bonferroni-adjusted significance *p* < 0.0025. (**b**) Comparison between clinical subgroups of total scores of the EA-QOL© questionnaire for children aged 8–17 years old, proxy-reports (left) and self-reports (right) between clinical subgroups. Only patients for whom both clinical data and total scores EA-QOL© were available were included in the analyses. ^A^ Only children ≥12 years old reported these items. Asterisks (*) indicate Bonferroni-adjusted significance *p* < 0.0025.

	2–7 Years Old (Proxy-Report)						8–17 Years Old (Proxy-Report)						8–17 Years Old (Self-Report)					
	Yes		No				Yes		No				Yes		No			
	*n*	Median (IQR)	*n*	Median (IQR)	*p*-Value	Effect Size (r)	*n*	Median (IQR)	*n*	Median (IQR)	*p*-Value	Effect Size (r)	*n*	Median (IQR)	*n*	Median (IQR)	*p*-Value	Effect Size (r)
**Surgical characteristics**																		
Primary repair	85	81.25 (68.30–92.42)	11	66.18 (46.88–76.47)	0.007	0.27	120	87.50 (80.43–93.46)	16	84.71 (65.70–89.84)	0.058	0.16	115	87.50 (79.17–95.46)	15	81.52 (68.48–90.63)	0.118	0.14
History of gastrostomy	15	66.18 (50.00–82.14)	82	82.03 (69.03–92.65)	0.003	−0.3	18	84.71 (65.84–89.06)	118	87.50 (80.43–93.55)	0.042 *	−0.17	17	82.29 (70.70–90.63)	113	87.50 (79.17–95.55)	0.114	−0.14
History of ≥1 dilatation	53	80.00 (67.16–92.52)	47	79.41 (67.31–87.50)	0.785	−0.03	68	88.35 (74.22–93.21)	68	87.20 (80.43–93.28)	0.689	−0.03	66	84.38 (73.95–95.01)	64	87.50 (81.32–94.23)	0.318	−0.09
**Digestive symptoms**																		
Heartburn	18	69.79 (60.00–81.99)	66	84.69 (72.52–93.08)	0.005	−0.31	17	83.70 (76.04–87.13)	114	87.83 (79.99–93.48)	0.08	−0.15	12	82.43 (80.73–88.72)	66	90.63 (79.30–96.63)	0.028	−0.25
Dysphagia ^A^	60	72.39 (62.68–85.88)	41	88.24 (77.99–94.85)	<0.001 *	−0.42	51	83.33 (72.92–90.63)	85	88.54 (83.33–95.61)	0.001 *	−0.29	39	81.25 (72.73–93.48)	44	91.49 (84.48–97.92)	0.001 *	−0.36
Vomiting	21	69.12 (61.65–78.24)	80	83.32 (70.21–92.65)	0.002 *	−0.31	6	73.08 (53.13–84.62)	130	87.50 (80.16–93.00)	0.167	−0.12	2	86.03	81	89.13 (78.27–95.83)	0.801	−0.03
**Respiratory symptoms**																		
Coughing	64	72.39 (62.68–85.88)	37	89.71 (78.31–95.71)	<0.001 *	−0.43	63	85.42 (73.96–91.30)	72	88.88 (82.95–95.13)	0.009	−0.22	42	85.10 (76.56–95.50)	41	90.63 (81.52–97.25)	0.078	−0.15
Wheezing	26	68.45 (58.46–80.35)	73	84.38 (71.45–93.02)	<0.001 *	−0.37	9	85.42 (74.40–91.06)	127	87.50 (80.21–93.42)	0.451	−0.06	11	81.25 (77.08–93.18)	72	89.68 (79.21–96.28)	0.181	−0.19
Dyspnea at rest	10	60.66 (44.28–72.93)	90	82.58 (69.03–92.30)	<0.001 *	−0.36	6	79.94 (72.66–88.94)	129	87.50 (79.78–93.48)	0.202	−0.11	7	79.17 (72.73–86.46)	75	89.58 (79.35–96.43)	0.106	−0.18
Dyspnea during physical activity	12	61.08 (46.09–68.53)	88	82.58 (70.55–92.06)	<0.001 *	−0.37	20	82.95 (72.06–87.50)	112	88.54 (80.26–93.75)	0.023	−0.2	25	80.68 (72.23–90.49)	57	91.30 (84.04–96.59)	0.002 *	−0.35
Chest tightness	3	45.31	84	82.81 (69.34–95.53)	0.041	−0.22	9	87.50 (78.46–91.74)	125	87.50 (79.78–93.48)	0.776	−0.02	14	81.97 (72.48–93.75)	68	89.58 (79.68–96.28)	0.089	−0.19
Airway infections	14	72.39 (57.58–82.26)	85	82.35 (68.30–92.65)	0.014	−0.25	12	72.40 (68.95–85.64)	123	88.16 (80.43–93.75)	0.002 *	−0.27	3	80.68	79	89.58 (79.67–95.83)	0.095	−0.18
**Feeding difficulties**																		
Avoiding certain food	35	66.18 (53.85–76.92)	61	84.39 (74.26–93.93)	<0.001 *	−0.54	31	80.68 (72.62–86.46)	104	88.54 (83.42–94.79)	<0.001 *	−0.33	12	79.17 (71.17–92.71)	69	89.58 (80.01–96.21)	0.064	−0.21
Eating small portions	43	70.00 (54.69–82.81)	55	85.94 (73.44–93.75)	<0.001 *	−0.45	26	78.65 (68.23–87.09)	108	88.88 (81.32–94.79)	<0.001 *	−0.37	13	73.96 (69.70–87.73)	69	90.63 (81.91–96.51)	0.001 *	−0.35
Energy-enriched food	21	58.82 (51.47–76.63)	79	83.83 (70.45–92.65)	<0.001 *	−0.42	8	70.55 (65.89–81.30)	128	87.50 (80.43–93.46)	0.005	−0.24	5	82.29 (71.33–90.63)	77	89.58 (79.17–95.83)	0.241	−0.13
Adjusted food consistency	25	67.65 (53.68–76.63)	73	85.29 (72.57–93.02)	<0.001 *	−0.44	1	80.43	134	87.50 (79.30–93.44)	0.426	−0.07	0		83	89.13 (79.17–95.83)	-	-
Needing >30 min to finish a meal	36	67.75 (55.35–79.89)	61	85.94 (75.74–93.93)	<0.001 *	−0.5	17	78.13 (69.27–84.69)	119	88.54 (80.68–93.75)	<0.001 *	−0.19	8	79.69 (73.44–85.50)	71	89.77 (79.35–96.59)	0.025	−0.25
Increased fluid intake during meals	45	75.00 (83.28–86.76)	51	85.00 (70.45–94.12)	0.015 *	−0.25	43	81.25 (71.88–86.46)	90	90.63 (83.24–95.50)	<0.001 *	−0.42	31	79.35 (71.74–90.63)	50	92.19 (84.21–97.92)	<0.001 *	−0.44
Nutrition through tube or gastrostomy	12	53.39 (45.70–61.10)	87	82.81 (70.45–92.65)	<0.001 *	−0.5	2	53.68	134	87.50 (80.00–83.44)	0.027	−0.19	1	68.75	82	89.36 (79.17–95.83)	0.138	−0.16
Adult support while eating	28	65.00 (50.74–78.96)	71	85.00 (72.06–92.65)	<0.001 *	−0.41	4	66.15 (54.69–76.26)	132	87.50 (80.43–93.46)	0.006	−0.24	0		83	89.13 (79.17–95.83)	-	-

## Data Availability

All dataset(s) supporting the conclusions of this article are available upon reasonable request.

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
