# Peer review of "Psychometric Performance of a Condition-Specific Quality-of-Life Instrument for Dutch Children Born with Esophageal Atresia"

_children, 2022, doi:10.3390/children9101508_

Round 1
Reviewer 1 Report
This study aims to assess the reliability and validity of Dutch version of QoL questionnaire for patients/families with esophageal atresia.
I only have a couple of comments:
Abstract: Please reformat for journal style: The abstract should be a single paragraph and should follow the style of structured abstracts, but without headings.
The authors have stated in results that the proportion of parents with high educational level was larger than that in the general population (58% vs. 36%). I think this could be one source of bias as well and should be added in discussion.
To summarize, the authors have done a great job with the translation and it appears to me that the questionnaire validation is carried out well. This work is definitely relevant for the Dutch population and especially families and children with esophageal atresia and the teams looking after them.
Author Response
Thank you for your valid comments
We have now changed the abstract according to the Journal style. We shortened the abstract and took out the headings.
The educational level of parents was already mentioned in our discussion, but we have tried to stress its importance further.
Reviewer 2 Report
The study is a translated Swedish protocol. The EA-QOL questionnaires have been translated and used in other European countries. Although some data have been generated, they suffer from limitations:
The response rates ranged from 39-51%.
Missing values were common.
What qualifies the questionnaires to be of good reliability and quality may need to be better defined for the conclusion to be drawn.
Author Response
Thank you for your comments, which we have tried to respond to below:
The response rates ranged from 39-51%.
Indeed, in the nationwide field test these were the response rates for the three different groups. Of course, one would always like the highest possible response rate. All patients were therefore reminded twice if they did not respond. We believe that sending more reminders would be unethical as patients may start to feel guilty if they do not want to participate. Despite this relatively low response rate we still had a large cohort of patients. We have added to the discussion to clearly stress the importance of this limitation.
Missing values were common.
Yes, this an important issue, but has likely to do with items by items not deemed applicable to an individual child’s situation by the responder. I refer to the discussion in which we comment on this.
What qualifies the questionnaires to be of good reliability and quality may need to be better defined for the conclusion to be drawn.
We did not define overall quality but did not conclude on that either. The reliability is defined in the methods section:
Internal reliability was considered good if Cronbach’s alpha ≥0.7 for the scales[32]. External reliability – both proxy-self and test-retest comparison – was evaluated using intra-class coefficients (ICCs), using a two-way random model, single measures and absolute agreement. It was considered poor (<0.50), moderate (0.50-0.74), good (0.75-0.90), or excellent (>0.90)[33].
Reviewer 3 Report
In the manuscript entitled " Psychometric performance of a condition-specific quality of life instrument for Dutch children born with esophageal atresia," Chantal A. ten Kate et al. assess the quality of life of children born with esophageal atresia with a condition-specific instrument.
In my opinion, the theme has a significant clinical interest, and the article contributes considerably.
The work is well written, and the Tables are clear and easy to understand. The Discussion provides relevant insights into the theme, and the study's limitations are well discussed.
Author Response
Thank you for your comments. They are well appreciated.
Round 2
Reviewer 2 Report
Great effort in running the questionnaires among the Dutch patient and parents. Detailed compilation and analysis of the data are evident.
In the study design, a clear set of criteria at the beginning to prove or disprove whether "The Dutch-translated EA-QOL© questionnaires showed good reliability and validity. "
may be helpful.
Author Response
Thank you for reviewing our manuscript once more. We are happy to hear found our adjustments helpful.
The criteria for good feasibility, reliability and validity have been described in detail under the subheading 'Statistical analysis'. To meet your suggestion, we have added the following sentence at row 47: 'following international guidelines regarding the criteria for feasibility, reliability and validity'.
Round 3
Reviewer 2 Report
Good effort in collection of data.